# Manifestation and Progression of Metabolic Dysfunction-Associated Steatotic Liver Disease in a Predominately African American Population at a Multi-Specialty Healthcare Organization

**DOI:** 10.3390/healthcare12151478

**Published:** 2024-07-25

**Authors:** Astha Saini, Brian Rutledge, Anirudh R. Damughatla, Mina Rasheed, Paul Naylor, Milton Mutchnick

**Affiliations:** Department of Internal Medicine, School of Medicine, Wayne State University, Detroit, MI 48201, USA; gi7518@wayne.edu (A.S.); drbrianrutledge@gmail.com (B.R.); hf9269@wayne.edu (A.R.D.); mrasheed@med.wayne.edu (M.R.); mmutchnick@med.wayne.edu (M.M.)

**Keywords:** MASLD, MASH, non-invasive serum markers, risk factors in NASH/NAFLD, racial disparity in NASH/NAFLD

## Abstract

African Americans (AA) have a high incidence of risk factors associated with MASLD (metabolic dysfunction-associated steatotic liver disease); the AA population has a lower incidence of MASLD and MASH (metabolic-associated steatotic hepatitis) than Caucasian and Hispanic Americans (non-AA). We investigated if underlying risk factor variation between AA and non-AA individuals could provide a rationale for the racial diversity seen in MASLD/MASH. Using ICD-10 codes, patients from 2017 to 2020 with MASLD/MASH were identified and confirmed to have either MASLD or MASH. Despite the large (>80%) AA population in our clinics, only 54% of the MASLD/MASH patients were African American. When the non-invasive NAFLD Fibrosis Scores (NFS) evaluated at early diagnosis were compared to the most recent values, the only increase in fibrosis score by NFS over time was in non-AA MASH patients. The increase in fibrosis only in non-AA MASLD patients is consistent with racial disparity in the disease progression in non-AA as compared to AA patients. Even with the large proportion of AA patients in our study, there was no significant racial disparity in the earliest assessment of either risk factors, laboratory values, or fibrosis scores that would account for racial disparity in the development and progression of MASLD.

## 1. Introduction

Non-alcoholic fatty liver disease (NAFLD) and non-alcoholic steatohepatitis (NASH) are increasingly prevalent chronic liver diseases in the United States [1,2,3,4,5,6]. In June 2023, a multi-society Delphi consensus statement introduced a new nomenclature—metabolic dysfunction-associated steatotic liver disease (MASLD), replacing the term NAFLD [7,8]. Obesity, type 2 diabetes mellitus (T2DM), hypertension (HTN), and hyperlipidemia contribute to steatosis, with at least one required to meet the revised definition, which includes “metabolic dysfunction” [9].

Despite a high incidence of risk factors among African American (AA) individuals, they have a lower incidence of steatosis and development of MASH compared to Caucasian and Hispanic Americans [10,11]. This racial disparity is also evident in mortality rates for MASLD/MASH cirrhosis and hepatocellular carcinoma (HCC), which are lower in non-Hispanic blacks compared to non-Hispanic and Hispanic whites. Patients with MASH represent a clinically at-risk subset of MASLD patients that continue to progress to advanced fibrosis and cirrhosis. The fibrosis stage of MASLD correlates with hepatic-related outcomes and influences clinical management.

Although biopsy was previously considered the definitive method to identify steatosis and steatohepatitis, there has been an increase in the use of non-invasive methods to stage the risk levels of MASLD/MASH patients [12,13,14]. Given the high number of AA patients in our clinic and the lack of data focused on MASLD/MASH in the AA population, we assessed the risk factors and non-invasive serum markers for differences between the AA and non-AA patients at their earliest clinic visits compared to their most recent visits. We also evaluated whether underlying risk factor variation could explain the racial diversity seen in MASLD/MASH. This longitudinal study assessed the characteristics of AA versus non-AA patients concerning the progression of liver fibrosis over time in an AA-predominant MASLD/MASH population.

## 2. Methods

We obtained medical records of adult patients with the ICD codes for NASH (K75.81; non-alcoholic steatohepatitis) or NAFLD (K76.0; fatty liver, not classified elsewhere) from 2017 to 2020 from the institutional electronic medical records (EMR) [15]. Institutional review board (IRB) approval was secured prior to data collection and analysis. We identified patients using ICD-10 codes for NAFLD/NASH, but for data analysis, we adhered to the use of at least one of the five metabolic dysfunction factors for the accurate identification of MASLD and MASH.

### 2.1. Data Collection

Demographic information, imaging findings, laboratory data, medical history, and pathology results were extracted from the EMR. Race was primarily self-identified, and patients were classified as African American (AA) or non-AA. Serum laboratory values were recorded at the earliest and most recent visits to monitor changes over time. Medical history, laboratory values, and medication use were used to define the presence of hypertension (BP 140/90 or medication therapy), hyperlipidemia (medication therapy), type 2 diabetes (HbA1c > 6.4, impaired fasting glucose, or medication therapy), and obesity (BMI > 25 kg/m^2^).

### 2.2. Definition of Liver Steatosis and Fibrosis

Liver steatosis was primarily defined by imaging (183/206 = 90%). Other categories included biopsy, cirrhosis by imaging or non-invasive scores in conjunction with metabolic factors in patients without other liver risk factors, and bariatric surgery (10%). Fibrosis was assessed using primarily biopsy findings, non-invasive lab-based calculations (NFS, FIB-4, APRI, BARD, Table 1, Figure 1) and/or imaging techniques (FibroScan, ultrasound, and/or CT). Two arbitrators (BR and PN) evaluated the data to confirm the diagnosis of MASLD (liver steatosis by imaging or BMI > 25 kg/m^2^ with minimal fibrosis [F0–F1]) or MASH (liver steatosis with significant fibrosis [F2–F4] or significant fibrosis/cirrhosis with metabolic risk factors and no other hepatitis risk factors).

### 2.3. Biopsy and Exclusion Criteria

Biopsy evaluation of MASLD or MASH was rare (7%), with reports including features such as macro vascular steatosis, ballooning degeneration of hepatocytes, scattered (mainly lobular) inflammation, apoptotic bodies, and Mallory–Denk bodies (MDBs). Patients were excluded from the MASLD/MASH study if they had competing liver disease risk factors (i.e., alcohol use disorder, chronic hepatitis C), insufficient information to confirm the diagnosis, or a history of any cancer. Additionally, patients who underwent bariatric surgery were excluded from the disease progression analysis.

### 2.4. Statistical Analysis

Statistical analysis was performed using the JMP-SAS version 17.2.0. Three statistical evaluations available in the program were used in this study. Analysis of variance (ANOVA) was used to compare the means of different groups of continuous variables. It was preferred over the *t*-test since it allowed for the comparison of variances for statistical significance across more than two groups simultaneously. Pearson’s chi-squared analysis was used for categorical variables to determine whether observed differences were statistically significant. It was chosen over the likelihood ratio test, as given the sample size, the results of both tests were identical. The paired *t*-test was applied to determine if the mean difference between two sets of data was significant, particularly when evaluating changes in the same individuals over time. Significant differences in the three tests were defined as a *p*-value less than 0.05.

## 3. Results

### 3.1. Correct Coding of Fatty Liver with or without Significant Fibrosis

Out of 282 ICD-10-coded NAFLD/NASH patients, we identified 189 patients with confirmed fatty liver and metabolic dysfunction (MASLD or MASH). Among the 84 patients with sufficient information to assess misdiagnosis, 40 patients (48%) had a documented history of significant alcohol consumption, 28 patients (33%) had chronic hepatitis C, 15 patients (18%) had no data confirming the diagnosis, and 1 patient had a hereditary disease (1%). When assessing racial disparity, AA patients were more likely to be misdiagnosed due to competing liver disease diagnoses (68% AA vs. 31% non-AA; *p* < 0.002).

Many overweight patients are treated with bariatric surgery which results in a loss of weight and has been demonstrated in a subset of patients to lead to MASLD improvement [20]. Table 2 presents a comparison between MASLD, MASH, and bariatric surgery patients with respect to their earliest medical record event. Bariatric surgery was infrequent among MASLD/MASH patients (16%). Bariatric surgery patients were more likely to be female and not have had a visit to see a gastroenterologist. These bariatric patients were included in the initial evaluation of MASLD/MASH patients to evaluate potential racial differences but not in the studies evaluating real world outcomes, given the presence of an intervention known to alter disease progression.

Of the 189 patients who met the criteria for either MASLD or MASH, 35 had ICD-10 codes for NAFLD but also had advanced fibrosis and met the criteria for a NASH ICD-10 code (Figure 2). These misdiagnosed patients with significant fibrosis represented half of the total number of patients with significant fibrosis in our dataset. In contrast, only 7 patients were incorrectly diagnosed with advanced fibrosis (ICD-10 for NASH) but did not have significant fibrosis and thus met the criteria for NAFLD. Overall, non-AA patients were more likely to be correctly diagnosed with NASH compared to AA patients (15/113 = 13% vs. 25/93 = 27%; *p* < 0.008).

### 3.2. Race, Liver Steatosis, and Fibrosis

Despite AA patients constituting over 80% of our medical center population, they made up only 55% of the MASLD/MASH patients (AA = 105, Asian = 12, Caucasian = 44, Hispanic = 12, Middle Eastern = 12, Other = 4). The prevalence of MASH (advanced fibrosis) at the earliest visit was similar in both AA and non-AA patients (33/103 = 32% vs. 39/93 = 42%; Table 3). In both the MASLD and MASH groups, the AA population trended towards a higher female ratio. Only 54% of patients had recorded visits to a gastroenterologist, suggesting that ICD-10 diagnoses were often made by non-GI physicians without subsequent referral to GI specialists. Based on the medical records, the primary indication for a referral to GI was elevated liver enzymes, although, at earliest visit, only 52% of patients had ALT/AST values of greater than 30 (unpublished results). The age of patients at diagnosis ranged from 15 to 81 years, with AA patients being slightly older (53 vs. 48 years; *p* = 0.023). MASH patients were older at the earliest date of service compared to MASLD patients (56 years vs. 48 years; *p* = 0.0002). Table 3 compares AA and non-AA MASLD/MASH patients across various characteristics at their earliest visit. Only APRI and FIB-4 scores showed racial differences (lower in AA vs. non-AA for both MASLD and MASH patients).

Common risk factors for MASLD/MASH, such as obesity (85%), diabetes (50%), hypertension (58%), and lipidemia (51%) were prevalent among MASLD/MASH patients. Metabolic syndrome (MtS), which includes HTN, DM2, BMI > 25 kg/m^2^, and lipidemia, was present in similar numbers in both MASH and MASLD patients [21]. There were no significant differences in the presence of HTN, DM, obesity, and MtS between AA and non-AA patients with either MASLD or MASH (Table 3). Lean MASLD/MASH (BMI < 25 kg/m^2^ with documented steatosis) was rare (9–16%).

Non-invasive serum-based markers (NFS, BARD, APRI, and FIB-4) were significantly higher in MASH patients compared to MASLD patients, regardless of race (Table 3). While there was overlap between the non-invasive serum assessments in MASLD and MASH patients, where FIB-4 and APRI were more specific than NFS (Table 3, Figure 3).

### 3.3. Race and Progression of Disease

Racial differences in disease progression of fibrosis were evaluated using three non-invasive markers (APRI, FIB-4, and NFS; Figure 4). The average time between assessments was 5.3 years for MASLD patients and 5.6 years for MASH patients. When non-invasive fibrosis assessments were compared temporally using matched-pair analysis, there was a significant increase in fibrosis progression as defined by NFS for the non-AA MASH patients (0.46 to 1.24; *p* = 0.004) compared to the AA MASH patients (0.39 to 0.45). Increased fibrosis in predominantly non-AA MASH patients was also observed with APRI (0.79 to 1.24) and FIB-4 (1.92 to 3.60). The only significant increase in fibrosis over time for AA individuals was defined by FIB-4 in the AA MASLD patients (0.74 to 0.95).

## 4. Discussion

At our predominantly African American urban academic medical center, only half of the patients with MAFLD/MASH were seen by a gastroenterologist (GI). Notably, a significant number of MASLD patients with advanced fibrosis were not identified as MASH based on ICD-10 codes. This misclassification may result in failures to refer these patients to GI specialists, potentially impacting the management of their advanced fibrosis and subsequent development of cirrhosis.

The proportion of MASLD and MASH patients identified as African American (AA) did not reflect the overall clinic population, where AA individuals represent over 80%. This lower proportion aligns with the existing literature suggesting a potential protective effect of possible genetic predilection to protection against MASLD/MASH [8,9,10,11]. Despite agreement across multiple studies, the underlying reasons for this racial disparity remain unclear and warrant further investigation. Additionally, our study in a large AA population found no significant racial differences in the presence of risk factors that could explain the lower incidence of MASLD/MASH among AA patients. The four primary risk factors (obesity, T2DM, lipidemia, and hypertension) were more prevalent in MASH patients than MASLD patients, regardless of race, with BMI > 25 kg/m^2^ being the most common risk factor.

Nearly half of the MASLD (39%) and MASH (41%) patients were not referred to a GI specialist, potentially reflecting a bias among primary care physicians that lifestyle counseling and managing metabolic risk factors are their responsibilities. This may also contribute to the heterogeneous evaluation and workup, including the ordering of imaging and FibroScans. While primary care management may suffice for MASLD patients, MASH patients, who have a higher risk of liver disease progression and require close monitoring for variceal bleeding and hepatocellular carcinoma, should ideally be referred to GI specialists. We anticipate an increase in GI referrals as therapeutic options for these conditions such as GLP-1, GIP, and resmetirom become more available [22,23,24].

Given the significant proportion of AA patients with MASLD/MASH in our study, we could evaluate racial diversity in disease progression using non-invasive fibrosis markers over an average period of five years. Matched-pair analysis revealed a significant increase in fibrosis progression, as defined by NFS, in non-AA NASH patients, whereas progression was less apparent in AA patients. This difference could reflect actual disease progression or the limitations of NFS in AA patients. Although APRI and FIB-4 showed similar trends, they did not achieve statistical significance.

MASLD patients exhibited only a modest increase in fibrosis across all three assessments, with FIB-4 being potentially useful for monitoring fibrosis progression in MASLD patients of both races. The lack of a significant increase in NFS among MASLD patients supports the notion that fibrosis progression accelerates once patients transition to MASH. Our results align with the observations that AA patients are less likely to experience progressive MASLD/MASH. Further investigation with a longer follow-up is necessary, particularly in the lower-risk MASLD population, to identify factors that may predispose patients to progress to MASH.

## 5. Conclusions

Our study found that despite the predominance of African American (AA) patients in our clinics, only 54% of the MASLD/MASH patients were AA. This lower percentage suggests a potential genetic protective effect against MASLD/MASH. Notably, due to the large AA clinic population, our study had a substantial number of AA patients, providing a robust basis for examining racial differences in disease progression.

When evaluating disease progression using non-invasive fibrosis markers, there was a significant increase in fibrosis progression as defined by the NFS in non-AA MASH patients, but not in AA MASH patients. This observed racial difference remains unexplained, as common risk factors such as obesity, diabetes mellitus (DM), hypertension (HTN), and metabolic syndrome (MtS) were not significantly different between AA and non-AA patients with MASLD or MASH.

Additionally, as in other studies, we identified a minority of individuals from both racial groups who exhibited steatosis as defined by imaging, despite lacking the three typical risk factors of obesity, T2DM, and HTN.

African genetic ancestry is associated with a lower frequency of the PNPLA3 allele, a variant which may be a risk factor for non-alcoholic fatty liver disease, suggesting a possible ancestry protective factor [24,25]. Further research is warranted to understand the underlying reasons for the observed racial disparity in MASLD/MASH progression. This may involve identifying additional risk factors or protective mechanisms specific to AA patients that could inform targeted interventions and improve disease management strategies for this population.

### Manuscript Highlights

Despite the predominance (>80%) of AA patients in the Wayne Health clinics, AA patients constituted only 54% of the MASLD/MASH patients. With respect to the presence of MASH in the total population of MASLD/MASH, there was no racial difference (AA = 34% vs. non-AA = 44%; *p* = 0.12).

Non-AA patients were more likely to be correctly diagnosed with MASH as compared to AA patients (14% vs. 28%; *p* < 0.0071).

When racial disparity was assessed, AA patients were more likely to be misdiagnosed due to competing liver disease diagnoses (AA = 68% vs. non-AA = 31%; *p* < 0.002).

HTN, hyperlipidemia, T2DM, BMI > 25 kg/m^2^, and metabolic syndrome were not significantly different when compared between the AA and non-AA patients with respect to both MASLD and MASH.

When the non-invasive fibrosis assessments were compared using matched-pair analysis, there was a significant increase in the progression of fibrosis as defined by NFS for the non-AA MASH patients (0.46 to 1.44, *p* = 0.004) as compared to the AA patients (−0.146 to 0.324).

## Figures and Tables

**Figure 1 healthcare-12-01478-f001:**
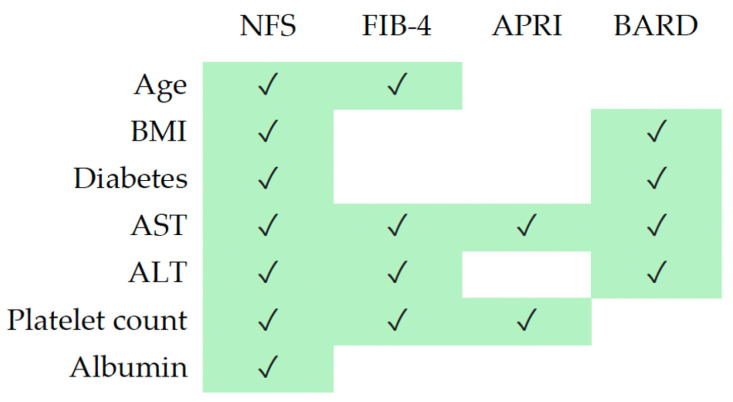
Criteria used for non-invasive testing of liver fibrosis.

**Figure 2 healthcare-12-01478-f002:**
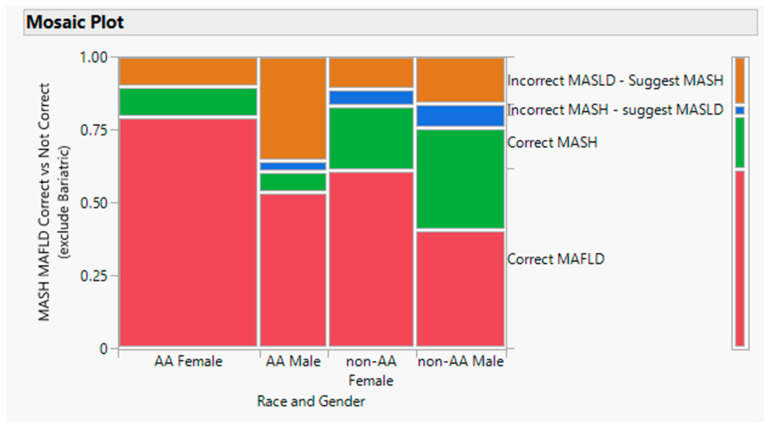
Ratio for MASLD to MASH by race and gender. The primary racial disparity was that MASH was incorrectly diagnosed primarily in AA males (*p* < 0.0009). Note that the diagnosis codes were based on the physician interpretation of NAFLD vs. NASH since the newly recommended nomenclature has not yet been adopted by the ICD-10 coding system.

**Figure 3 healthcare-12-01478-f003:**
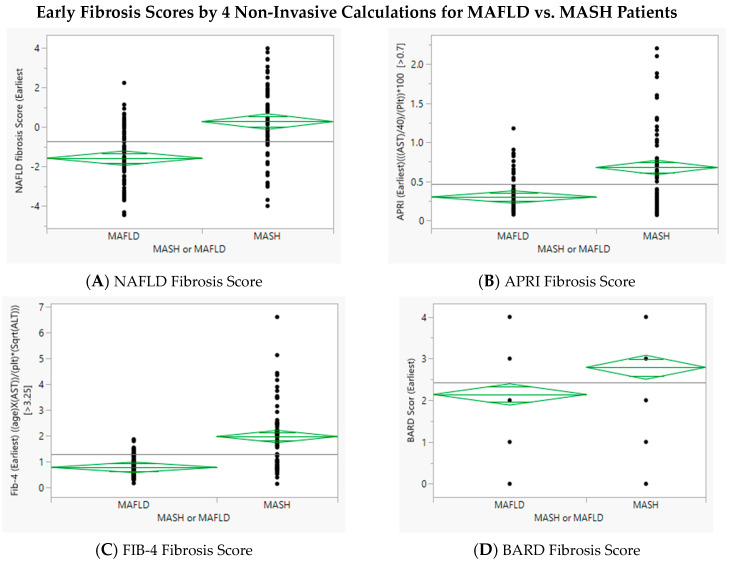
Four non-invasive markers of fibrosis were evaluated for utility in differentiating between patients with steatosis and minimal fibrosis (MASLD) and those with significant fibrosis (MASH). With respect to average, all four were greater in MASH as compared to MASLD but there was significant overlap with respect to individual values.

**Figure 4 healthcare-12-01478-f004:**
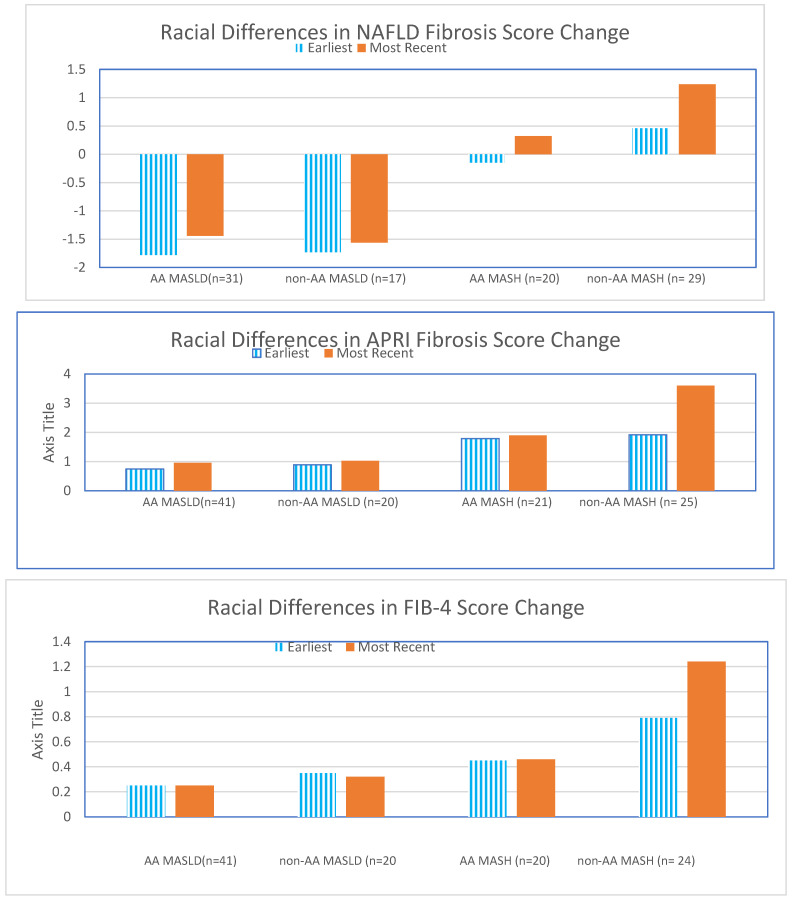
Change in fibrosis by three non-invasive assessment in MASLD/MASH patients. NAFLD Fibrosis Score increased over time in MASLD/MASH patients. Non-AA MASH patients had the largest increase over time (*p* = 0.004 using pair-wise analysis). The racial difference in fibrosis progression in MASH patients was also found in APRI and FIB-4 fibrosis.

**Table 1 healthcare-12-01478-t001:** Formulas for non-invasive lab-based calculations.

NAFLD Fibrosis Score (NFS) [16]	−1.675 + (0.037 × age [years]) + (0.094 × BMI [kg/m^2^]) + (1.13 × IFG/diabetes [yes = 1, no = 0]) + (0.99 × AST/ALT ratio) − (0.013 × platelet count [×109/L]) − (0.66 × albumin [g/dL])		
Fibrosis-4 Index for Liver Fibrosis (FIB-4) [17]	(Age* × AST)/(Platelets x √(ALT)) * = Use with caution in patients <35 or >65 years old, as the score has been shown to be less reliable in these patients.		
AST to Platelet Ratio Index (APRI) [18]	(AST in IU/L)/(AST Upper Limit of Normal in IU/L)/(Platelets in 109/L)		
BARD Score for NAFLD Fibrosis [19]	Variable		Points
	BMI ≥ 28	No	0
		Yes	1
	AST/ALT ratio ≥ 0.8	No	0
		Yes	2
	Diabetes	No	0
		Yes	1

**Table 2 healthcare-12-01478-t002:** Differences between MASLD, MASH, and bariatric surgery populations.

Total Patients (n = 189)	MASLD (n = 106)	MASH (n = 56)	Bariatric Surgery (n = 27)	*p* Value Overall (Pearson)	*p* Value MASLD vs. MASH
Race (% AA) (n = 189)	58%	46%	56%	0.33	0.14
Gender (%M) (n = 189)	33%	51%	4%	0.0001	0.02
GI visit (% GI) (n = 189)	61%	59%	22%	0.0008	0.76
BMI ≥ 25 kg/m^2^ (n = 185)	84%	91%	100%	0.01	0.23
Type 2 Diabetes (n = 189)	41%	59%	63%	0.03	0.03
Lipidemia (n = 140) ^1^	69%	66%	67%	0.92	0.70
Hypertension (n = 189) ^2^	54%	66%	70%	0.19	0.16
Metabolic Syndrome ^3^	27%	32%	41%	0.34	0.47
Lean MASLD/MASH ^4^	16%	9%	0%	0.06	0.23
Risk Factor (Average Number Exclude)	2.6	2.8	2.9	0.32	0.07

^1^ Lipidemia (Triglycerides ≥ 150/HDL Female ≤ 50; Male ≤ 40 or on medication). ^2^ Hypertension (HTN; 140/90 or on medication. ^3^ Metabolic Syndrome (HTN, DM2, BMI > 25 kg/m^2^, Lipidemia). ^4^ Lean MASLD/MASH (BMI < 25 kg/m^2^ but a fatty liver and at least one other risk factor).

**Table 3 healthcare-12-01478-t003:** Comparison of MASLD and MASH patients by race at early diagnosis.

Total Population (n = 189)	MASLD (n = 120)			MASH (n = 69)		
	AA (n = 70)	non-AA (n = 50)	*p*-Value	AA (n = 33)	non-AA (n = 36)	*p*-Value
Gender (% Male)	24%	69%	0.16	33%	53%	0.10
GI Diagnosing (%GI)	60%	50%	0.28	55%	53%	0.88
HTN	63%	50%	0.16	70%	61%	0.45
Diabetes	44%	40%	0.64	55%	67%	0.30
Obesity (BMI > 25 kg/m^2^)	84%	89%	0.42	91%	94%	0.57
% Lean MASLD/MASH	16%	11%	0.41	9%	6%	0.57
Complete Metabolic Syndrome (HTN/DM2/BMI > 25/TriG ≥ 150 mg/dL/HDL F ≤ 50 mg/dL; M ≤ 40 mg/dL)	28%	28%	0.96	33%	36%	0.81

	MASLD			MASH		
Early Diagnosis Calculated Scores (AA vs. non-AA)	AA	Non-AA		AA	Non-AA	
NFS (>0.67 high fibrosis)	−1.38	−1.84	0.18	−0.16	0.39	0.23
BARD (<2 high fibrosis)	2.14	2.10	0.87	2.76	2.73	0.79
APRI (>1.5 high fibrosis)	0.25	0.38	0.007	0.50	0.84	0.01
FIB-4 (>3.25 high fibrosis)	0.73	0.89	0.042	1.77	2.13	0.32

	AA			Non-AA		
Early Diagnosis Calculated Scores (MASLD vs. MASH)	MASLD	MASH		MASLD	MASH	
NFS	−1.38	−0.16	0.002	−1.84	0.399	0.0001
BARD	2.14	2.75	0.04	2.10	2.73	0.04
APRI	0.25	0.50	0.0001	0.38	0.84	0.0002
FIB-4	0.73	1.77	0.0001	0.89	2.13	0.0001

Note: Bariatric surgery patients were included in the early demographics study.

## Data Availability

The data used in this study were from the Wayne Health Electronic Medical Records and the dataset is available with the appropriate approval of the WSU IRB.

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
