# Peer review of "Manifestation and Progression of Metabolic Dysfunction-Associated Steatotic Liver Disease in a Predominately African American Population at a Multi-Specialty Healthcare Organization"

_healthcare, 2024, doi:10.3390/healthcare12151478_

Round 1

Reviewer 1 Report

Comments and Suggestions for Authors

I appreciate the chance to review the article titled "Manifestation and Progression of Metabolic Dysfunction Associated Steatotic Liver Disease in a Predominately African American Population at a Multi-Specialty Healthcare Organization".

The article presents an intriguing issue of racial diversity in the epidemiology of metabolic dysfunction associated with steatotic liver disease.

The "Introduction" section is concise, but adequately introduces the reader to the topic of the article.

Please refer to the reviewer's comments:

1. The authors introduced non-invasive markers (NFS, FIB-4, APRI, BARD) for assessing liver fibrosis in lines 72-73. Subject experts possess knowledge of these issues and the equations necessary for their calculation, however, it is worth presenting this information collectively, such as in a tabular style. This will also allow for the expansion of the information provided in lines 62-63 concerning the data acquired from laboratory test results.

2. In lines 82-83, the authors highlighted that patients with additional risk factors for liver disease were not included in the study. However, it would be more precise to explicitly mention these risk factors, especially considering that they were eventually presented in Section 3.1 as well.

3. The available data about the statistical analysis methods (lines 86-89) are insufficient. Please provide further details, such as information on the distribution of variables and the selection of statistical tests.

4. Please pay attention to typos, such as in table 1 where "BM>25" should be corrected to "BMI>25". Additionally, it is important to include the units of measurement (kg/m2), as this omission is consistently observed throughout the text.

5. In Figure 1, it is worth presenting, in addition to the column color presentation, numerical data - I suggest modifying the figure in this respect, it will become more readable;

6. While the text already contains this information, it is advisable to include the p value in figures such as Figure 2 or Figure 3. This will enhance the clarity of the graphs and make their interpretation easier. In the case of Figure 3, similar to the previous point, it would be beneficial to incorporate numerical data directly into the graph.

7. I suggest introducing a collective list of abbreviations used in the manuscript.

I would be delighted to review the revised version of the manuscript. I congratulate the authors on the work done and wish you good luck in your future scientific activities.

Author Response

Comment 1: The authors introduced non-invasive markers (NFS, FIB-4, APRI, BARD) for assessing liver fibrosis in lines 72-73. Subject experts possess knowledge of these issues and the equations necessary for their calculation, however, it is worth presenting this information collectively, such as in a tabular style. This will also allow for the expansion of the information provided in lines 62-63 concerning the data acquired from laboratory test results.

Response 1: Thank you for your comment. We have added a figure to present this information. Please see Figure 1. The previous figures have been renumbered.  

Comment 2: In lines 82-83, the authors highlighted that patients with additional risk factors for liver disease were not included in the study. However, it would be more precise to explicitly mention these risk factors, especially considering that they were eventually presented in Section 3.1 as well.

Response 2: Thank you for your comment. We had added the following to line 82-83: “Patients were excluded from the MASLD/MASH study if they had competing liver disease risk factors (ie: alcohol use disorder, chronic hepatitis C), insufficient information to confirm the diagnosis, or a history of any cancer.”  

Comment 3: The available data about the statistical analysis methods (lines 86-89) are insufficient. Please provide further details, such as information on the distribution of variables and the selection of statistical tests.

Response 3: Thank you for your comment. We have added the following: "Statistical analysis was performed using JMP-SAS software. Three statistical evaluations available in the program were used in this study. Analysis of variance (ANOVA)  was used to compare variances across the means of groups of different groups and was continuous variables.  It was preferred over the t-test since it provides the option of comparing variance for statistical significance across more than 2 groups at a time. Pearson chi-square analysis was used for categorical variables, to determine whether observed differences are statistically significant. It was chosen over the likelihood ratio although given the size of the samples, the two are identical in their results.   Matched pairs t-test determines if the mean difference between two sets of data is significant and is applied when evaluating changes in the same individuals over time.  Significant differences in the three tests were defined as a p value less than 0.05.

Comment 4: Please pay attention to typos, such as in table 1 where "BM>25" should be corrected to "BMI>25". Additionally, it is important to include the units of measurement (kg/m2), as this omission is consistently observed throughout the text.

Response 4: Thank you for this correction. We have addressed all typos and added units of measurement.  

Comment 5: In Figure 1, it is worth presenting, in addition to the column color presentation, numerical data - I suggest modifying the figure in this respect, it will become more readable.

Response 5: Thank you for your comment. When we add numerical data to the figure, the image appears cluttered. We opted to detail the numerical values in the text to avoid a busy image. If you still recommend added numerical data, please let us know and we can redesign this figure.  

Comment 6: While the text already contains this information, it is advisable to include the p value in figures such as Figure 2 or Figure 3. This will enhance the clarity of the graphs and make their interpretation easier. In the case of Figure 3, similar to the previous point, it would be beneficial to incorporate numerical data directly into the graph.

Response 6: Thank you for your comment. When we add numerical data to the figure, the image appears cluttered. We opted to detail the numerical values in the text to avoid a busy image. If you still recommend added numerical data, please let us know and we can redesign this figure. 

Comment 7: I suggest introducing a collective list of abbreviations used in the manuscript.

Response 7: Thank you for your comment. We have added a list of abbreviations.  

Reviewer 2 Report

Comments and Suggestions for Authors

The manuscript entitled " Manifestation and Progression of Metabolic Dysfunction Associated Steatotic Liver Disease in a Predominately African American Population at a Multi-Specialty Healthcare Organization’’ examines the racial disparities in the incidence and progression of MASLD and MASH among African American (AA) and non-AA populations. Despite a high incidence of risk factors among AA individuals, they have a lower incidence of MASLD and MASH compared to Caucasian and Hispanic Americans. The study uses ICD-10 codes to identify patients with MASLD/MASH and evaluates non-invasive serum markers and risk factors. Risk factors such as obesity, diabetes, and hypertension are highly prevalent in the AA population. However, this high prevalence does not correspond to a proportionally higher incidence of MASLD/MASH. Addressing these disparities is crucial for developing effective prevention and treatment strategies tailored to specific populations.

There are specific points needs to be considered:

Specific points

·       One critical point is the absence of references for the ICD codes utilized in the analysis. Additionally, the definitions of hypertension, hyperlipidemia, type 2 diabetes (T2D), obesity and liver steatosis require appropriate citations.

·       In Table 1: Lipidemia was defined by TG≥150 and HDL Female ≤50;Male ≤40. How about LDL levels? please provide appropriate citation for this as well. Also, Metabolic Syndrome definition needs a reference.

·       The discussion section in this paper is relatively short and lacks a comprehensive analysis of their findings in relation to recent work in the literature. It does not adequately compare or contrast their results with those of other studies, which limits the depth and context of their conclusions. By not situating their findings within the broader scope of existing research, the authors miss the opportunity to highlight the significance and implications of their work. For instance, Variants in genes such as PNPLA3 and TM6SF2, which are associated with liver fat content and inflammation, may exhibit different frequencies and effects among African Americans (PMID: 35710086). The authors could refer to this paper and may be others as a potential explanation for the lower incidence of MASL/MASH. A more thorough discussion, incorporating recent literature, would enhance the relevance of their findings.

Author Response

1. One critical point is the absence of references for the ICD codes utilized in the analysis. Additionally, the definitions of hypertension, hyperlipidemia, type 2 diabetes (T2D), obesity and liver steatosis require appropriate citations.

Thank you for your comment. We have added a reference for the ICD codes (reference 14), definitions for hypertension, hyperlipidemia, T2DM, obesity and liver steatosis (reference 8).  

2. In Table 1: Lipidemia was defined by TG≥150 and HDL Female ≤50; Male ≤40. How about LDL levels? Please provide appropriate citation for this as well. Also, Metabolic Syndrome definition needs a reference. 

Thank you for your comment. The AASLD diagnostic criteria for MASLD includes only BMI, DM/serum glucose, blood pressure, TG and HDL, LDL is not part of the criteria. Therefore, we have not included the LDL levels (PMID: 37363821). We have added a reference for metabolic syndrome (reference 17) 

3. The discussion section in this paper is relatively short and lacks a comprehensive analysis of their findings in relation to recent work in the literature. It does not adequately compare or contrast their results with those of other studies, which limits the depth and context of their conclusions. By not situating their findings within the broader scope of existing research, the authors miss the opportunity to highlight the significance and implications of their work. For instance, Variants in genes such as PNPLA3 and TM6SF2, which are associated with liver fat content and inflammation, may exhibit different frequencies and effects among African Americans (PMID: 35710086). The authors could refer to this paper and maybe others as a potential explanation for the lower incidence of MASL/MASH. A more thorough discussion, incorporating recent literature, would enhance the relevance of their findings. 

Thank you for your comment and reference. After reading the paper by Cavalcante et al, we have added to our conclusion section. We also added another reference to suggest ancestry-mediated protection against MASLD.  

Round 2

Reviewer 1 Report

Comments and Suggestions for Authors

The authors responded to most of the comments, but not all of them (including the one regarding statistical analysis) - I suggest adding appropriate information in accordance with the recommendation from the previous review; this is crucial for the possibility of using the results of this study in the future work of other research groups.

Regarding my comment about the characterization of non-invasive markers, the presented form is interesting (although the description for Figure 1 is incorrect - age, BMI or diabetes can hardly be called laboratory tests). However, I recommend using precise equations for these markers to enhance the comprehensiveness of the information - it is also essential to cite the original works by the authors of these formulas. You have the possibility to keep the current format and include suitable formulas, such as above the table. Please consider this change.

I am content with the remaining responses.

Author Response

Comment: Regarding my comment about the characterization of non-invasive markers, the presented form is interesting (although the description for Figure 1 is incorrect - age, BMI or diabetes can hardly be called laboratory tests). However, I recommend using precise equations for these markers to enhance the comprehensiveness of the information - it is also essential to cite the original works by the authors of these formulas. You have the possibility to keep the current format and include suitable formulas, such as above the table. Please consider this change.

Response: Thank you for your comment. We have added Table 1 with all formulas for non-invasive tests. References have been added. We also added to the Statistical Analysis section. All changes have been highlighted in yellow.